# One Patch, One Text: Sparse Alignment for Closing CLIP's Modality Gap for Compositional Zero-Shot Learning

## Abstract

Compositional Zero-Shot Learning (CZSL) aims to recognize unseen attribute-object compositions with learned primitives (*e.g.*, attribute and object) knowledge from seen compositions. Previous methods achieve remarkable results by leveraging powerful cross-modal alignment capabilities of CLIP. However, they largely ignore inherent limitations arising from information-imbalanced image-text training data, notably the modality gap. In this work, we propose SAC, a novel Sparse Alignment framework to effectively Close CLIP's modality gap for CZSL. Specifically, we conduct ***sparse alignment*** between textual representations and their semantically relevant visual patches, which reduces redundant visual information and mitigates information imbalance within image-text pairs. Subsequently, leveraging the reduced visual information of this alignment, the ***visual adaptive condensation*** module is guided to adaptively condense critical visual cues into a unified representation. Finally, we introduce a ***dynamically updated memory bank*** that stores samples from both seen and unseen compositions (drawn from historical test data). This design bypasses the modality gap by relying solely on visual classification, while simultaneously improving generalization to unseen compositions. Experiments on three benchmarks demonstrate that our method gains significant improvements over a strong CLIP-based method under closed-world and open-world settings.

## 1 Introduction

Humans can recognize novel concepts by recombining familiar components. For instance, even without prior exposure, a *black swan* can be identified by combining the concepts of *swan* and *black* (Naeem et al., 2021; Lake, 2014). Compositional Zero-Shot Learning (CZSL) (Misra et al., 2017; Nagarajan & Grauman, 2018; Li et al., 2020; Karthik et al., 2021; Khan et al., 2023) aims to endow models with this ability: Given training on seen attribute–object pairs (e.g., white *swan*, *black* cat), the model must generalize to unseen compositions (e.g., *black swan*) by disentangling and recombining primitives. Recent CZSL approaches build on the powerful visual–semantic space of CLIP (Radford et al., 2021), pre-trained on large-scale image–text pairs. To improve the recognition of attribute–object compositions, techniques have explored enhanced visual–text alignment (Lu et al., 2023), primitive disentanglement (Huang et al., 2024), priority calibration (Li et al., 2024), and semantic mining (Wu et al., 2025). While these methods achieve strong performance, they still inherit fundamental limitations of CLIP. In particular, the modality gap caused by *information imbalance* (Schrodi et al., 2025) remains unresolved.

The *modality gap* (Liang et al., 2022) refers to the geometric separation of image and text embeddings in the visual–semantic space of CLIP. As a result, image embeddings tend to be similarly close to many text categories, rather than tightly aligned with their corresponding descriptions (Zhang et al., 2023). With limited matched-pair supervision, the model overemphasizes negative pairs, pushing image and text embeddings into disjoint regions of the space. Research findings have attributed the modality gap to contrastive loss (Liang et al., 2022) or temperature parameter (Udandarao, 2022; Shi et al., 2023), but recent evidence points to a deeper cause: *information imbalance* in training image–text pairs (Schrodi et al., 2025). Captions usually describe only salient objects, whereas **images encode richer details**. This mismatch weakens the supervision signal from

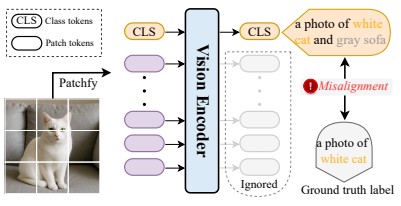 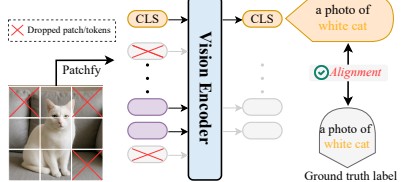

(a) Illustration of information imbalance | (b) Visual reduction by token drop.

Table 1: AUC and modality gap on C-GQC dataset.

| # Tokens | AUC |
|---|---|
| Full$_{(AUC)}$ | 14.3 |
| Drop$_{(AUC)}$ | 14.7 |
| Δ | +0.4 |
| Full$_{(gap)}$ | 0.15 |
| Drop$_{(gap)}$ | 0.14 |
| Δ | -0.01 |

Figure 1: a) shows that the vision side encodes excessive information relative to the ground-truth text label; b) demonstrates a simple vision-token reduction to align the information from both modalities better.

matched pairs, as illustrated in Fig. 1a. The imbalance is especially harmful in fine-grained tasks such as CZSL, where salient attributes can easily be contaminated by the surrounding context.

We observe that prior works typically discard patch tokens and rely solely on the CLS token (Fig. 1a), which aggregates all visual information and often includes redundancy. We hypothesize that the imbalance can be mitigated if less information is forced into the CLS token. To validate this, we conduct a pilot experiment (see Sec. 3): finetuning CLIP with LoRA (Zanella & Ben Ayed, 2024) (Full) while randomly dropping patch tokens at the input (Drop). As shown in Fig. 1b and Tab. 1, even this naive variant yields notable AUC gains and gap reduction on C-GQA (Naeem et al., 2021).

Building on the observation that reducing visual context improves balance across modalities, we propose a principled *three-stage* framework, SAC (Sparse Alignment for Closing modality gap). Prior methods rely on the CLS token as the visual representation, which encodes global information and often includes irrelevant context, while output patch tokens are simply discarded (Fig. 1a). Inspired by CLIP-based segmentation methods (Chi et al., 2025; Liang et al., 2023) that use patch tokens at the output for local prediction, we instead exploit patch tokens to restore image–text balance in CZSL by suppressing irrelevant visual information. In **Stage I**, for each composition text prompt, we align it with the most semantically congruent patch token (via Max operation), yielding a sparse set of selected patches. This information-balanced training quickly adapts CLIP to attribute–object recognition as the two modalities become better aligned. However, naive rule-based token selection is overly rigid, since some relevant patches may be discarded. We therefore introduce **Stage II**, a *Visual Adaptive Condensation* (VAC) module that consolidates all patch features into a unified representation. VAC is guided by soft labels from Stage I, enabling it to recover contextually relevant information while remaining regulated by the reduced visual signals from Stage I. Finally, in **Stage III**, we maintain a dynamic memory bank that stores high-confidence visual representations from VAC. During inference, the memory is updated with predictions on both seen and unseen compositions. This provides visual classification references *across* instances and accelerates adaptation to unseen compositions, further mitigating the modality gap in cross-modal alignment.

The main contributions are summarized as follows: 1) We design a ***Sparse Alignment*** (SA) training paradigm tailored to the characteristics of CZSL downstream tasks, effectively mitigating modality gap issue. 2) We propose the ***Visual Adaptive Condensation*** (VAC) module to adaptively excavate and compress critical visual information into a consolidate representation, preserve the information balance between image-text pairs. 3) We devise a ***Dynamically Updated Memory Bank*** that bypasses modality gap issue by provides pure-visual modality classification references while accelerating adaptation to unseen compositions. 4) Extensive experiments demonstrate significant improvements over existing state-of-the-art methods across popular benchmarks.

## 2 RELATED WORK

**Compositional Zero-Shot Learning** aims to recognize unseen attribute–object compositions by exploiting knowledge of attributes and objects learned from seen compositions. Existing methods largely follow two paradigms. The first (Misra et al., 2017; Naeem et al., 2021; Nagarajan & Grauman, 2018; Li et al., 2020; Liu et al., 2023) treats each composition as a single entity and aligns it directly with the corresponding attribute–object word embedding (*e.g.*, via graph learning, gated networks, or transformations). The second (Hao et al., 2023; Kim et al., 2023; Yang et al., 2020; Li et al., 2022; Wang et al., 2023b) disentangles attributes and objects, training separate classifiers

to enhance compositional reasoning. Recently, Vision–Language Models (VLMs) have been increasingly adopted, leveraging their strong cross-modal alignment and broad knowledge to advance CZSL performance (Nayak et al., 2023; Huang et al., 2024; Li et al., 2024; Lu et al., 2023; Bao et al., 2023; Zheng et al., 2024; Wu et al., 2025; Qu et al., 2025). However, current approaches often rely on CLIP alignment capabilities while neglecting its inherent modality gap, which limits performance, particularly in fine-grained CZSL. We show that strategically reducing visual information can substantially mitigate this gap, leading to improvements beyond prior work.

**Modality Gap** was first identified in (Liang et al., 2022), which described it as a geometric phenomenon in which embeddings of different modalities occupy disjoint regions of the shared embedding space of CLIP. The mismatches between image-text pairs push all visual and text embeddings away from each other, and cause image embeddings to tend to be similarly close to text prompts of many categories, rather than tightly aligned with their corresponding descriptions (Zhang et al., 2023). Subsequent studies (e.g., (Udandarao, 2022; Shi et al., 2023)) suggested that the temperature parameter can partially regulate this gap. More recently, (Schrodi et al., 2025) provided a rigorous analysis, linking the gap to information imbalance in image–text pairs, where the model faces an alignment–consistency trade-off. Importantly, the study showed that balancing information reduces both the modality gap and model uncertainty, leading to consistent performance gains. Inspired by these findings, we selectively reduce visual information to align with textual data in the CZSL task, building an information-balanced paradigm that mitigates the modality gap.

**Retrieval-Augmented Models** were first introduced in NLP, where external memory is integrated into inference phase via retrieval mechanisms. Recently, (Zhang et al., 2022) employs a $k$-nearest neighbor classifier to boost classification without fine-tuning. (Udandarao et al., 2023) retrieves samples from external or generated sources. (Rong et al., 2023) leverages retrieved prompts to refine CLIP representations, while (Jing et al., 2024) builds a database of primitive knowledge from training data. However, these databases are typically static and lack access to unseen compositions, which limits their effectiveness in CZSL. Therefore, we propose a dynamically updated memory bank that archives seen samples and selectively incorporates reliable test instances during inference.

## 3 PRELIMINARIES

**Problem Formulation.** Compositional zero-shot learning (CZSL) aims at learning a model from seen compositions to recognize unseen compositions that share the same primitives. Given an attribute set $\mathcal{A} = \{a^1, a^2, \ldots, a^{|\mathcal{A}|}\}$ and an object set $\mathcal{O} = \{o^1, o^2, \ldots, o^{|\mathcal{O}|}\}$, the composition set is defined as $\mathcal{C} = \{c^1, c^2, \ldots, c^{|\mathcal{C}|}\}$, where each composition is composed of attribute and object $c = (a, o)$. The composition set $\mathcal{C}$ is divided into disjoint seen composition set $\mathcal{C}_s$ and the unseen composition set $\mathcal{C}_u$, where $\mathcal{C}_s \cap \mathcal{C}_u = \varnothing$ and $\mathcal{C}_s \cup \mathcal{C}_u = \mathcal{C}$. According to the standard generalized zero-shot learning (Naeem et al., 2021), model can only access images in $\mathcal{C}_s$ during training, while the testing set $\mathcal{C}$ contains both seen and unseen compositions. Here, we can define the training set as $\mathcal{D}_{tr} = \{(x, c)|x \in \mathcal{X}_{tr}, c \in \mathcal{C}_s\}$ and testing set as $\mathcal{D}_{ts} = \{(x, c)|x \in \mathcal{X}_{ts}, c \in \mathcal{C}\}$. In the closed-world setting, only the known compositions (all compositions in the dataset) are considered. In the open-world setting (Karthik et al., 2022; Liu et al., 2023), the test set contains all possible compositions, i.e., $\mathcal{C} = \mathcal{A} \times \mathcal{O}$, which is a more challenging setting.

**Visual Representations**. Given an input image $x \in \mathbb{R}^{H \times W \times 3}$, visual encoder $\phi_{\text{vis}}$ of CLIP outputs a sequence of visual tokens (features) denoted as $\boldsymbol{V} = [\boldsymbol{v}_{\text{CLS}}, \boldsymbol{v}_1, \ldots, \boldsymbol{v}_L] \in \mathbb{R}^{(L+1) \times D}$, where $[\boldsymbol{v}_1, \ldots, \boldsymbol{v}_L]$ are $L$ patch tokens and $\boldsymbol{v}_{\text{CLS}}$ is the class token which aggregates global information.

**Textual Representations**. Prevailing approaches employ learnable soft prompts tailored to attributes, objects, and their compositions to generate textual representations. Specifically, the prompts are denoted as $\boldsymbol{\theta}_a = [\boldsymbol{p}_a^1, \ldots, \boldsymbol{p}_a^m, \boldsymbol{w}_a]$, $\boldsymbol{\theta}_o = [\boldsymbol{p}_o^1, \ldots, \boldsymbol{p}_o^m, \boldsymbol{w}_o]$ and $\boldsymbol{\theta}_c = [\boldsymbol{p}_c^1, \ldots, \boldsymbol{p}_c^m, \boldsymbol{w}_a, \boldsymbol{w}_o]$, respectively. Among them, "*a photo of*" is used to initialize $\boldsymbol{p}_a^{1:m}$, $\boldsymbol{p}_o^{1:m}$ and $\boldsymbol{p}_c^{1:m}$, and $w_a$ and $w_o$ are vocabulary of the attribute and object. These soft prompts are then fed into text encoder $\psi_{\text{text}}$ of CLIP to generate textual representations, denoted as $\boldsymbol{t}_a$, $\boldsymbol{t}_o$ and $\boldsymbol{t}_c$.

**Standard Cross-Modal Alignment.** Prevailing CZSL methods conventionally employ the CLIP native alignment mechanism to calculate the probability using the `[CLS]` token. Here, $\tau$ is the temperature parameter of pre-trained CLIP. All representations in the prediction process are $l2$-

normalized, and we omit the notation for simplicity:

$$p(c^i|x) = \frac{\exp(\boldsymbol{v}_{\text{cls}} \cdot \boldsymbol{t}_c^i / \tau)}{\sum_{k=1}^{|\mathcal{C}_s|} \exp(\boldsymbol{v}_{\text{cls}} \cdot \boldsymbol{t}_c^k / \tau)}, \tag{1}$$

**Motivations.** Most prior works adopt the cross-modal alignment in Eq. 1, overlooking CLIP's key limitation: the modality gap (Liang et al., 2022), which stems from *information imbalance* (Schrodi et al., 2025) between visual ($v_{\text{CLS}}$) and textual ($t_c$) representations. As shown in Fig. 1a, the global visual representation $v_{\text{CLS}}$ often carries redundant context beyond its textual label.

We argue that this imbalance can be alleviated by suppressing excess visual information. A naive approach is to randomly drop input patch tokens, so that less visual information is aggregated to $v_{\text{CLS}}$. To validate this, we extend the pilot experiment in Fig. 1b and Tab. 1 by varying the drop rate, and measure the modality gap using Relative Modality Gap (RMG) (Schrodi et al., 2025). As shown in Fig. 2 (Top), high drop rates discard too much visual content and degrade performance, whereas moderate rates improve both accuracy and alignment. Notably, AUC gains coincide with reduced RMG, confirming that controlled *vision reduction* is an effective strategy for modality balance.

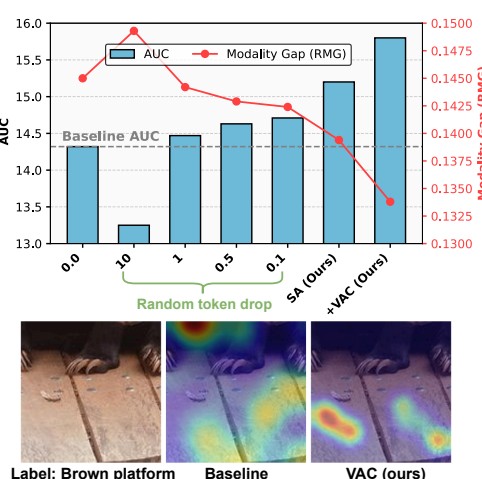

Motivated by these observations, we aim to reduce redundant visual content in a principled way. Random token dropping, however, removes useful information, disrupts semantic relations, and requires manual threshold tuning. Inspired by CLIP-based segmentation methods (Chi et al., 2025; Liang et al., 2023), where patch tokens represent local regions, we retain all patches during processing and modulate visual information at the output through token-level representations.

Figure 2: **Top:** Observation between modality gap and performance (AUC) on C-GQA dataset. We randomly drop patches with a certain probability (%) at the input, achieving a straightforward form of visual information reduction. **Bottom:** and example of attention visualization.

We introduce SAC, a framework that selectively reduces visual information while preserving critical semantics to restore balance in CZSL. As shown in Fig. 2 (top), our *Sparse Alignment* (SA) and *Visual Adaptive Condensation* (VAC) improve performance while narrowing the modality gap. Fig. 2 (bottom) further illustrates the effect: the baseline $v_{\text{CLS}}$ over-attends to both the `Brown bear` and the `Brown platform`, since they share the **"Brown"** attribute, whereas our VAC focuses only on the correct `Brown platform` leading to more sparse attention.

## 4 METHODS

### 4.1 SPARSE ALIGNMENT

Instead of relying on the global $v_{\text{CLS}}$ token, we shift to local representations by modulating the full set of output features $\boldsymbol{V} \in \mathbb{R}^{(L+1)\times D}$. We compute similarity scores between all tokens in $\boldsymbol{V}$ and the textual representations of seen compositions $\boldsymbol{T}_c \in \mathbb{R}^{|\mathcal{C}_s|\times D}$, and each composition $c$ select the patch token with the highest semantic relevance $\boldsymbol{s}_c$ via a $\max(\cdot)$ operation:

$$\boldsymbol{s}^c = \max_{l=1}^{L+1} \boldsymbol{S}^{l,c}, \quad \text{for } c = 1, 2, \ldots, |\mathcal{C}_s|, \qquad \text{where } \boldsymbol{S} = \boldsymbol{V}\boldsymbol{T}_c^{\top}, \tag{2}$$

with $\boldsymbol{S} \in \mathbb{R}^{(L+1)\times|\mathcal{C}_s|}$ denoting patch-to-composition similarities, and all representations are $\ell_2$-normalized. The intuition is that each patch token encodes a local region, and selecting the most informative token for each textual representation preserves discriminative features while suppressing redundant visual information. As only a subset of patches is retained, this process yields a

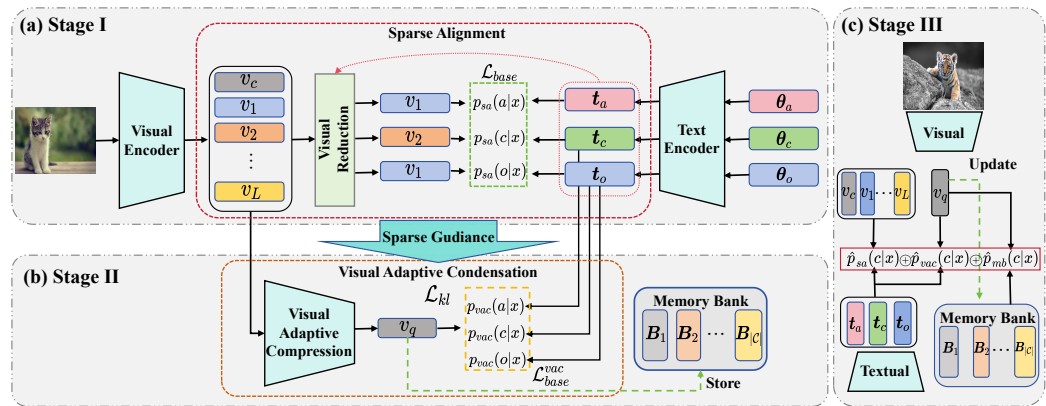

Figure 3: **The overview of SAC.** (a) **Stage I :** We perform *Sparse Alignment* between patch and text representation by reducing redundant visual information, detailed in Fig. 7. (b) **Stage II :** With the sparse guidance from sparse alignment, *Visual Adaptive Condensation* is learned to adaptively condense critical visual cues and stores condensed representations of seen compositions in a *Memory Bank*. (c) **Stage III :** During inference, we dynamically update the bank with reliable test samples from all compositions, and combine predictions from three modules for final results.

sparse alignment between visual and textual representations, which we denote as **Stage I: Sparse Alignment (SA)**. The resulting similarity scores $s \in \mathbb{R}^{|\mathcal{C}_s|}$ for all compositions are then used for reformulating the learning objective as follows:

$$\mathcal{L}_c = -\frac{1}{|\mathcal{D}_{tr}|} \sum_{x \in \mathcal{D}_{tr}} \log p_{sa}(c^i|x), \quad p_{sa}(c^i|x) = \frac{\exp(s^i/\tau)}{\sum_{k=1}^{|\mathcal{C}_s|} \exp(s^k/\tau)}, \quad (3)$$

Furthermore, we directly align textual representations of primitives with visual representations *without explicit disentanglement*, thus fully preserving the powerful cross-modal alignment capability of CLIP. Specifically, the learning objectives for primitives ($\mathcal{L}_a$ and $\mathcal{L}_o$) are similar to compositions, by replacing $T_c$ in Eq.2 with $T_a$ or $T_o$ followed by Eq. 3. The base training loss is defined as:

$$\mathcal{L}_{base} = \mathcal{L}_c + \mathcal{L}_a + \mathcal{L}_o. \quad (4)$$

To assess the role of patch tokens, we blend predictions from SA and the [CLS] token as $(1 - W) \times \text{SA} + W \times \text{[CLS]}$. As Fig.4 (left) shows, reducing the contribution of patch tokens in sparse alignment clearly decreases performance, indicating that the [CLS] token introduces excess visual information that contaminates alignment.

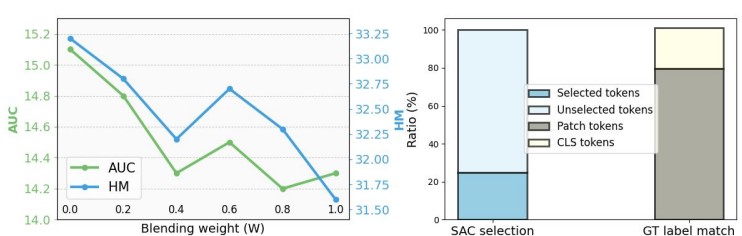

Figure 4: **Left:** Performance trend of C-GQA when SA is blended with the [CLS] token. **Right:** Ratio of patches selected by SA and proportion of top-response tokens that match the GT of C-GQA.

Furthermore, in Fig.4 (right), we measure the proportion of patches exhibiting high class-specific activation (blue bars), which constitute only ∼25% of all tokens, and empirically confirm the sparsity of the alignment. We also calculate the proportion of samples where the highest-response tokens matching the ground-truth label come from patch tokens (gray bar) rather than the [CLS] token in C-GQA. Overall, these results validate that adaptive, patch-level operation effectively alleviates information imbalance and addresses the modality gap.

### 4.2 VISUAL ADAPTIVE CONDENSATION

Although SA effectively reduces visual information to establish an information balance in the training data, such visual reduction may discard semantically valuable cues that are crucial for model

learning. To mitigate this limitation, we introduce **Stage II: Visual Adaptive Condensation (VAC)**, which adaptively aggregates essential information across all patches. This design compensates for potential information loss introduced by SA while preserving the benefits.

Specifically, VAC employs a learnable query embedding $q \in \mathbb{R}^{1 \times D}$ together with a stack of $K$ processing blocks. The query embedding dynamically attends to and aggregates semantically important information from all tokens in $V$, without being restricted to any single patch. Each block incorporates multi-head cross-attention mechanisms (Vaswani et al., 2017) and feed-forward network (FFN) (Liu et al., 2021) to enable adaptive information extraction and refinement. The condensed representation produced by the query embedding, denoted as $v_q$, is then used for prediction, with the learning objective for VAC formulated as:

$$\mathcal{L}_c^{vac} = -\frac{1}{|\mathcal{D}_{tr}|} \sum_{x \in \mathcal{D}_{tr}} \log p_{vac}(c^i|x), \quad p_{vac}(c^i|x) = \frac{\exp(v_q \cdot t_c^i/\tau)}{\sum_{k=1}^{|\mathcal{C}_s|} \exp(v_q \cdot t_c^k/\tau)}, \tag{5}$$

Similar to SA, we also utilize $v_q$ for primitives prediction without representation disentanglement. The prediction and learning objective ($\mathcal{L}_a^{vac}$ and $\mathcal{L}_o^{vac}$) formulae follow a similar form to Equation 5 and are omitted here for brevity. We denote the base learning objective for VAC module as follows:

$$\mathcal{L}_{base}^{vac} = \mathcal{L}_c^{vac} + \mathcal{L}_a^{vac} + \mathcal{L}_o^{vac}. \tag{6}$$

To ensure that condensed representation $v_q$ captures crucial semantic information while suppressing redundant visual details, we introduce a distillation objective (Hinton et al., 2015; Touvron et al., 2021) that preserves the reduced visual signal in SA, which is formally defined as:

$$\mathcal{L}_{kl} = -\frac{1}{|\mathcal{D}_{tr}|} \sum_{x \in \mathcal{D}_{tr}} p_{vac} \log(\frac{p_{vac}}{p_{sa}}), \tag{7}$$

where $p_{sa}$ and $p_{vac}$ denote the composition predictions produced by SA and VAC, respectively. By minimizing the Kullback-Leibler divergence (Kullback & Leibler, 1951) between these two probabilistic distributions, the distillation objective encourages $v_q$ to adaptively retain the critical visual information preserved by SA. The overall loss for VAC can be denoted as:

$$\mathcal{L}_{vac} = (1 - \alpha) \cdot \mathcal{L}_{base}^{vac} + \alpha \cdot \mathcal{L}_{kl}, \tag{8}$$

where $\alpha$ is a weight coefficient that balances the contribution of distillation loss.

## 4.3 DYNAMICALLY UPDATED MEMORY BANK

The above modules partially alleviate the modality gap. To further address this challenge, we propose **Stage III: Dynamically Updated Memory Bank**, which stores the condensed visual representations of all compositions. For each input sample, visual prototypes are retrieved from this memory bank. By relying exclusively on visual-modal representations for prediction, the memory bank effectively bypasses the modality gap and enhances recognition performance.

Specifically, we construct a memory bank (Jing et al., 2024; Zhang et al., 2022) $\mathbf{B} \in \mathbb{R}^{|\mathcal{C}| \times N \times D}$, where $N$ is the memory size for each composition, $\mathcal{C}$ denotes the total number of compositions. High-confidence samples from VAC are dynamically select to update the bank according to:

$$\mathbf{B}_{i,j} = v_q, \quad \text{if } \arg\max p_{vac} = i \text{ and } H(p_{vac}) < T_{i,j}, \tag{9}$$

where $i$ and $j$ represent the $j$-th stored sample in the $i$-th composition, $H(\cdot)$ is the entropy (Shannon, 1948) of the prediction. Notably, the $j$-th stored sample has the highest entropy $T_{i,j}$ in the $i$-th composition. This update process replaces the stored sample with the highest entropy with the current sample. During inference, the memory bank is continuously updated with test samples.

Next, we formalize the classification procedure based on the memory bank. For each input sample $v_q$, prototypes $\mathbf{P}$ of all compositions are retrieved from the memory bank through a query operation, formulated as:

$$p_{mb}(c^i|x) = \frac{\exp(v_q \cdot \mathbf{p}^i/\tau)}{\sum_{k=1}^{|\mathcal{C}|} \exp(v_q \cdot \mathbf{p}^k/\tau)}, \quad c^i = \text{softmax}((v_q \cdot \mathbf{B}_{i,:}^T)/\tau_{mb})\mathbf{B}_{i,:}, \tag{10}$$

where $\tau_{mb}$ is a temperature coefficient during retrieval which is empirically set to 0.1 for all datasets. $\mathbf{p}^i$ indicates the sample-adaptive prototype of the $i$-th composition, $\mathbf{B}_{i,:}$ are the stored samples of $i$-th composition. Notably, the memory bank operates in a training-free manner and performs prediction solely within the visual modality. To enable inference on unseen compositions when no visual samples are available, we initially insert the textual representations into memory bank.

Table 2: **The experimental results on closed-world settings.**

| Method | UT-Zappos | | | | MIT-States | | | | C-GQA | | | |
|---|---|---|---|---|---|---|---|---|---|---|---|---|
| | S | U | HM | AUC | S | U | HM | AUC | S | U | HM | AUC |
| CLIP[ICML'21] (Radford et al., 2021) | 15.8 | 49.1 | 15.6 | 5.0 | 30.2 | 46.0 | 26.1 | 11.0 | 7.5 | 25.0 | 8.6 | 1.4 |
| CoOp[IJCV'22] (Zhou et al., 2022) | 52.1 | 49.3 | 34.6 | 18.8 | 34.4 | 47.6 | 29.8 | 13.5 | 20.5 | 26.8 | 17.1 | 4.4 |
| CSP[ICLR'23] (Nayak et al., 2023) | 64.2 | 66.2 | 46.6 | 33.0 | 46.6 | 49.9 | 36.3 | 19.4 | 28.8 | 26.8 | 20.6 | 6.2 |
| DFSP[CVPR'23] (Lu et al., 2023) | 66.7 | 71.7 | 47.2 | 36.0 | 46.9 | 52.0 | 37.3 | 20.6 | 38.2 | 32.0 | 27.1 | 10.5 |
| PLID[ECCV'24] (Bao et al., 2023) | 67.3 | 68.8 | 52.4 | 38.7 | 49.7 | 52.4 | 39.0 | 22.1 | 38.8 | 33.0 | 27.9 | 11.0 |
| CDS[CVPR'24] (Li et al., 2024) | 63.9 | 74.8 | 52.7 | 39.5 | 50.3 | 52.9 | 39.2 | 22.4 | 38.3 | 34.2 | 28.1 | 11.1 |
| Troika[CVPR'24] (Huang et al., 2024) | 66.8 | 73.8 | 54.6 | 41.7 | 49.0 | 53.0 | 39.3 | 22.1 | 41.0 | 35.7 | 29.4 | 12.4 |
| LogiCzsl[CVPR'25] (Wu et al., 2025) | 69.6 | 74.9 | 57.8 | 45.8 | 50.8 | 53.9 | 40.5 | 23.4 | 44.4 | 39.4 | 33.3 | 15.3 |
| ClusPro[ICLR'25] (Qu et al., 2025) | 70.7 | 76.0 | 58.5 | 46.6 | 52.1 | **54.0** | 40.7 | 23.8 | 44.3 | 37.8 | 32.8 | 14.9 |
| **SAC** | **73.3** | **76.8** | **62.0** | **50.0** | **53.2** | 53.0 | **40.8** | **24.0** | **45.8** | **39.5** | **34.8** | **16.2** |

### 4.4 Training and Inference

**Training**. The overall learning objective is derived from the losses in SA (Eq. 4) and VAC (Eq. 8) modules, and the detailed training scheme can be found in supplementary material §D:

$$\mathcal{L} = \mathcal{L}_{base} + \mathcal{L}_{vac}. \tag{11}$$

**Inference**. The final prediction is derived by integrating the outputs from the three modules mentioned above, and the overall process can be formulated as follows:

$$\hat{p}(c|x) = \beta \cdot \bar{p}_{sa}(c|x) + (1 - \beta) \cdot (\bar{p}_{vac}(c|x) + \gamma \cdot p_{mb}(c|x)), \tag{12}$$

where $\bar{p}_{sa}(c|x)$ and $\bar{p}_{vac}(c|x)$ can be decomposed into the following terms:

$$\bar{p}(c|x) = p(c|x) + p(a|x) \cdot p(o|x). \tag{13}$$

## 5 Experiments

### 5.1 Experimental settings

**Datasets.** We conduct experiments on three benchmarks: fine-grained shoes UT-Zappos (Yu & Grauman, 2014), natural image MIT-States (Isola et al., 2015) and C-GQA (Naeem et al., 2021).

**Evaluation Metric.** Following generalized CZSL methods (Naeem et al., 2021; Liu et al., 2023), our method is evaluated on both seen and unseen compositions. Here, we report four metrics, such as Seen Accuracy (S), Unseen Accuracy (U), Harmonic Mean (HM) and Area Under Curve (AUC).

**Implement Details.** We adopt pre-trained CLIP ViT-L/14 model (Radford et al., 2021) as our backbone. Following previous works (Qu et al., 2025), we tune the visual encoder of CLIP with LoRA (Zanella & Ben Ayed, 2024). The detailed experimental settings are in supplementary material §A.

### 5.2 Comparison to State-of-the-Arts

In this section, we compare our SAC with several state-of-the-art (SOTA) approaches across three widely-used benchmarks under both closed-world setting Tab. 2 and open-world setting Tab. 3. In closed-world setting, our proposed SAC achieves the best performance on almost all metrics across all three datasets. Specifically, our SAC achieves the best AUC of 50.0, 24.0, and 16.2, along with the best HM of 62.0, 40.8, and 34.8 across the three datasets. Moreover, our proposed SAC attains the highest seen accuracy (S) and nearly the best unseen accuracy (U) across all three datasets. In open-world setting, our proposed SAC still outperforms the top-leading methods across all three datasets. To be specific, our SAC achieves the state-of-the-art AUC and HM with 41.7 (54.8), 9.3 (23.0) and 4.6 (15.3) on three datasets. According to seen accuracy and unseen accuracy, our method still outperforms other competing methods. In summary, the comprehensive results demonstrate that our proposed Sparse Alignment training framework effectively mitigates the modality gap, exhibiting strong robustness across datasets and settings. Comparison with more methods can be found in supplementary material §B.

Table 3: **The experimental results on open-world settings.**

| Method | UT-Zappos | | | | MIT-States | | | | C-GQA | | | |
|---|---|---|---|---|---|---|---|---|---|---|---|---|
| | S | U | HM | AUC | S | U | HM | AUC | S | U | HM | AUC |
| CLIP[ICML'21] (Radford et al., 2021) | 15.7 | 20.6 | 11.2 | 2.2 | 30.1 | 14.3 | 12.8 | 3.0 | 7.5 | 4.6 | 4.0 | 0.3 |
| CoOp[IJCV'22] (Zhou et al., 2022) | 52.1 | 31.5 | 28.9 | 13.2 | 34.6 | 9.3 | 12.3 | 2.8 | 21.0 | 4.6 | 5.5 | 0.7 |
| CSP[ICLR'23] (Nayak et al., 2023) | 64.1 | 44.1 | 38.9 | 22.7 | 46.3 | 15.7 | 17.4 | 5.7 | 28.7 | 5.2 | 6.9 | 1.2 |
| DFSP[CVPR'23] (Lu et al., 2023) | 66.8 | 60.0 | 44.0 | 30.3 | 47.5 | 18.5 | 19.3 | 6.8 | 38.3 | 7.2 | 10.4 | 2.4 |
| PLID[ECCV'24] (Bao et al., 2023) | 67.6 | 55.5 | 46.6 | 30.8 | 49.1 | 18.7 | 20.0 | 7.3 | 39.1 | 7.5 | 10.6 | 2.5 |
| CDS[CVPR'24] (Li et al., 2024) | 64.7 | 61.3 | 48.2 | 32.3 | 49.4 | 21.8 | 22.1 | 8.5 | 37.6 | 8.2 | 11.6 | 2.7 |
| Troika[CVPR'24] (Huang et al., 2024) | 66.4 | 61.2 | 47.8 | 33.0 | 48.8 | 18.7 | 20.1 | 7.2 | 40.8 | 7.9 | 10.9 | 2.7 |
| LogiCzsl[CVPR'25] (Wu et al., 2025) | 69.6 | 63.7 | 50.8 | 36.2 | 50.7 | 21.4 | 22.4 | 8.7 | 43.7 | 9.3 | 12.6 | 3.4 |
| ClusPro[ICLR'25] (Qu et al., 2025) | 71.0 | 66.2 | 54.1 | 39.5 | 51.2 | **22.1** | 23.0 | 9.3 | 41.6 | 8.3 | 11.6 | 3.0 |
| **SAC** | **72.9** | **66.7** | **54.8** | **42.3** | **52.9** | 21.2 | **23.1** | **9.4** | **45.5** | **11.5** | **15.3** | **4.6** |

Table 4: **Ablation studies on main innovations.**

| | UT-Zappos | | | | MIT-States | | | | C-GQA | | | |
|---|---|---|---|---|---|---|---|---|---|---|---|---|
| | S | U | HM | AUC | S | U | HM | AUC | S | U | HM | AUC |
| *Baseline* | 66.7 | 69.8 | 51.0 | 37.5 | 48.0 | 52.5 | 37.9 | 21.2 | 44.2 | 37.2 | 31.6 | 14.3 |
| $+\mathcal{L}_{base}$ | 67.6 | 75.5 | 56.6 | 44.4 | 50.3 | 51.9 | 39.1 | 22.0 | 44.6 | 38.5 | 33.4 | 15.2 |
| $+\mathcal{L}_{base}^{q}$ | 70.3 | 75.7 | 58.1 | 46.2 | 50.4 | 52.3 | 39.2 | 22.2 | 45.4 | 38.5 | 34.0 | 15.6 |
| $+\mathcal{L}_{kl}$ | 72.1 | 76.4 | 60.2 | 48.6 | 51.5 | 52.6 | 39.6 | 23.0 | 45.5 | 38.9 | 34.3 | 15.8 |
| *+Memory Bank* | 72.9 | 76.4 | 61.4 | 49.3 | 52.9 | 52.6 | 40.1 | 23.4 | 45.7 | 38.9 | 34.6 | 16.0 |
| *+Dynamically Update* | **73.3** | **76.8** | **62.0** | **50.0** | **53.2** | **53.0** | **40.8** | **24.0** | **45.8** | **39.5** | **34.8** | **16.2** |

## 5.3 ABLATION STUDY

**Main components.** We conduct a deep investigation of the key components in our SAC, including Sparse Alignment (SA), Visual Adaptive Condensation (VAC) and Dynamically Updated Memory Bank. As shown in Tab. 4, the experiments are conducted as follows: 1) $\mathcal{L}_{base}$ applies the SA training paradigm to mitigate the modality gap. 2) $\mathcal{L}_{base}^{q}$ trains VAC with a conventional cross-entropy objective. 3) $\mathcal{L}_{kl}$ leverages the reduced visual signal guide the VAC module to adaptively extract critical visual information. 4) *Memory Bank* stores samples only for seen compositions. 5) *Dynamically Update* stored samples for seen and unseen compositions during inference simultaneously. As we can see in Tab. 4, SA ($\mathcal{L}_{base}$) establishes an information-balanced training paradigm and yields significant improvements across three datasets. VAC ($\mathcal{L}_{base}^{q}$ and $\mathcal{L}_{kl}$) guided by the reduced visual information in SA, further adaptively excavates critical visual information, leading to additional performance improvements. *Memory Bank* bypasses the modality gap by providing visual-modality classification references, improving seen accuracy. Finally, *Dynamically Update* enhances both seen and unseen accuracy by continuously enriching the memory bank during inference. More detailed ablation study can be found in supplementary material §B.

Table 5: **Visual Reductions.**

| | S | U | HM | AUC |
|---|---|---|---|---|
| mean | 44.2 | 35.7 | 32.0 | 13.9 |
| attention | **44.9** | 34.3 | 30.9 | 13.5 |
| linear | 44.6 | 36.8 | 32.5 | 14.5 |
| **max** | 44.6 | **38.5** | **33.4** | **15.2** |

Table 6: **SA in Troika.**

| | S | U | HM | AUC |
|---|---|---|---|---|
| w/o SA | 41.0 | 35.7 | 29.4 | 12.4 |
| w SA | **41.9** | **36.1** | **30.7** | **13.1** |
| w/o SA | 66.8 | **73.8** | 54.6 | 41.7 |
| w SA | **70.0** | 72.9 | **55.2** | **43.1** |

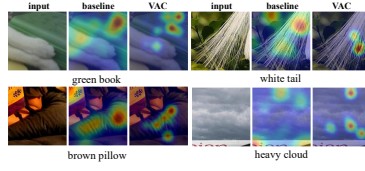

Figure 5: **Visualization of VAC.**

**Visual Reduction in Sparse Alignment.** In Sparse Alignment, we leverage a simple yet effective max operation to achieve visual information suppression. Here, we conduct experiments on different operations, *e.g.*, mean pooling, learnable patch-specific weight (multi-head attention or linear layer) in Tab. 5. Mean pooling cannot focus on critical patches, as it produces a uniform contribution from

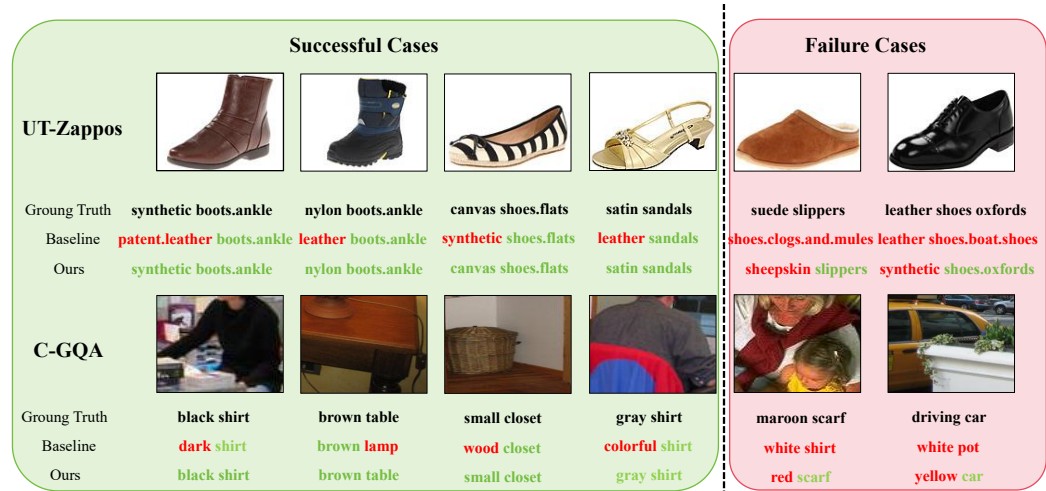

Figure 6: **Qualitative results in UT-Zappos and C-GQA.** Visual comparison between baseline and our method SAC. Green indicates the right prediction and red denotes the wrong prediction.

all patches. The generalization ability of learnable patch-specific weights on unseen combinations is poor, while the max operation achieves the best performance.

**Improvements of Sparse Alignment in Troika.** The sparse alignment we proposed is a straightforward plug-and-play training paradigm. To validate its effectiveness, we integrate this module into SOTA method Troika (Huang et al., 2024) for comprehensive evaluation. As shown in Tab. 6, sparse alignment does not introduce additional learnable parameters, and improves AUC from 12.4 to 13.1 in C-GQA (1-2nd rows) and from 41.7 to 43.1 in UT-Zappos (3-4th rows), respectively.

**Visualization of Visual Adaptive Condensation.** To qualitatively evaluate whether the VAC module indeed exploits the critical patches, we visualize the attention weights of VAC and baseline (last layer of CLIP) in Fig. 5. As we can see, the VAC module adaptively excavates critical visual information while minimizing attention to redundant details, such as "$cat$" in "$green\ book$", "$leaf$" in "$white\ tail$" and "$word$" in "$heavy\ cloud$". More visualization results are in §C.

## 5.4 QUALITATIVE RESULTS

**Qualitative results.** We present qualitative results of UT-Zappos and C-GQA datasets in Fig. 6, including both successful and failure cases compared with the baseline. As we can see, SAC effectively attends to the primary object without being distracted by redundant visual cues, such as "$brown\ lamp$" vs. "$brown\ table$" or "$colorful\ shirt$" vs. "$gray\ shirt$". In failure cases, although the model may misclassify a composition, it usually captures at least one primitive correctly. These errors often arise from semantic similarity (e.g., "$red$" vs. "$maroon$", "$suede$" vs. "$sheepskin$") or from heavy entanglement of multiple attributes and objects, which obscures the distinction between relevant and redundant information and leads to incorrect predictions. More qualitative results can be found in supplementary material §C.

## 6 CONCLUSION

In this paper, we propose SAC, a sparse alignment framework to mitigate the modality gap in CLIP for CZSL. Through the analysis of the modality gap, we propose an Sparse Alignment (SA) training paradigm that establishes an information balance between image-text pairs by suppressing redundant visual information, thereby mitigating the modality gap. Subsequently, we introduce a Visual Adaptive Condensation (VAC) module, which adaptively captures critical visual information guided by SA and compensates for potential information loss in SA at the same time. Finally, we design a memory bank dynamically updated during inference to perform classification within the visual modality, bypassing the modality gap. We hope that our work can inspire future research on mitigating modality gap in compositional learning.

## REPRODUCIBILITY STATEMENT

For a fair comparison, we use the Troika (Huang et al., 2024) and RAPR (Jing et al., 2024) codebase. The pre-trained CLIP models are directly sourced from the OpenAI CLIP repository. Other components, such as feed-forward network and multi-head attention, are standard PyTorch functions. The detailed hyper-parameters and training scheme are reported in supplementary material. The trained model and full code will be released upon publication of the paper.

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

This appendix provides additional details for the ICLR 2026, titled "One Patch, One Text: Sparse Alignment for Closing CLIP's Modality Gap for Compositional Zero-Shot Learning". It is orgnized as follows:

- §A Detailed Experiment Settings.
- §B More Ablation Experiments.
- §C More Qualitative Experiments.
- §D Pseudo-code.
- §E Statement for Using Large Language Models.

## A DETAILED EXPERIMENT SETTINGS

**Detailed Dataset Split Statistics.** We conduct experiments on three widely-used datasets: UT-Zappos, MIT-States, and C-GQA. UT-Zappos is a fine-grained dataset composed of 50,025 shoes images with 16 attributes ($e.g.$, Cotton, Nylon), 12 objects ($e.g.$, Shoes.Heels, Boots.Ankle) and 116 compositions. MIT-States contains 53,753 natural images with 115 attributes ($e.g.$, Ancient, Broken), 245 objects ($e.g.$, Computer, Tree) and 1962 compositions. C-GQA is the most extensive dataset containing 39,298 images with 453 attributes, 870 objects and more than 9,500 compositions. Following the standard split, we divide the compositions into *train / validation / test* splits. The detailed splits are shown in Tab. 7. $|\mathcal{C}_s|$ indicates the number of seen compositions, $|\mathcal{C}_u|$ is the number of unseen compositions, $\mathcal{X}$ represents the number of samples in the corresponding splits.

Table 7: Detail of data split statistics.

| Dataset | Compositions | | | Train | | Val | | Test | |
|---|---|---|---|---|---|---|---|---|---|
| | $|\mathcal{A}|$ | $|\mathcal{O}|$ | $|\mathcal{A}| \times |\mathcal{O}|$ | $|\mathcal{C}_s|$ | $|\mathcal{X}|$ | $|\mathcal{C}_s|/|\mathcal{C}_u|$ | $|\mathcal{X}|$ | $|\mathcal{C}_s|/|\mathcal{C}_u|$ | $|\mathcal{X}|$ |
| UT-Zappos | 16 | 12 | 192 | 83 | 22998 | 15 / 15 | 3214 | 18 / 18 | 2914 |
| MIT-States | 115 | 245 | 28175 | 1262 | 30338 | 300 / 300 | 10420 | 400 / 400 | 12995 |
| C-GQA | 413 | 674 | 278362 | 5592 | 26920 | 1252 / 1040 | 7280 | 888 / 923 | 5098 |

**Detailed Evaluation Metrics.** Following the generalized CZSL evaluation protocol (Naeem et al., 2021; Liu et al., 2023), our method is evaluated on both seen and unseen compositions. We report the four widely used metrics for a comprehensive evaluation. Seen Accuracy (S) and Unseen Accuracy (U) are computed to evaluate the best classification performance on seen and unseen compositions. Using Seen Accuracy as $x$-axis and Unseen Accuracy as $y$-axis, we calibrate a bias scalar (Naeem et al., 2021) on Unseen Accuracy and obtain a seen-unseen accuracy curve. Then, we compute and report the Area Under the Curve (AUC). Meanwhile, we compute the best Harmonic Mean (HM) between Seen Accuracy and Unseen Accuracy at a specific bias scalar.

**More Implementation Details.** For network initialization, we load the weights of CLIP (Radford et al., 2021) and tune the image encoder with LoRA (Zanella & Ben Ayed, 2024). The *Sparse Alignment* suppresses semantically irrelevant regions to achieve information balance in image-text pairs. The overall pipeline of *Sparse Alignment* is illustrated in Fig. 7. The *Visual Adaptive condensation* module is implemented with $K$ blocks composed of multi-head attention and feed-forward network. The number of blocks $K$ is set to 3, 3 and 1 for UT-Zappos, MIT-States and C-GQA, respectively. The *Dynamically Updated Memory Bank* does not introduce additional parameters, as the retrieval and prediction processes are calculated on the condensed visual representations without transformation. The coefficient $\alpha$ for distillation loss in Eq. 8 is set to 0.5, 0.9 and 0.5 for three datasets. The coefficient $\beta$ in Eq. 12 is set to 0.3, 0.7 and 0.7. The coefficient $\gamma$ in Eq. 12 is set to 0.5, 0.4 and 0.1. For the number of stored samples in *Dynamically Updated Memory Bank* is set to 16, 24 and 16. We train our model for 15, 10 and 15 epochs with Adam Optimizer (Kingma & Ba, 2014). The learning rates are initialized at $2e-4$, $5e-5$ and $5e-4$, where the learning rate is scheduled by the StepLR (PyTorch, 2025). During training, we set batch size to 64, 64 and 16 for three datasets. All the experiments are conducted on a single NVIDIA RTX 3090 GPU. More ablation experiments on hyper-parameters is presented in Sec. B.

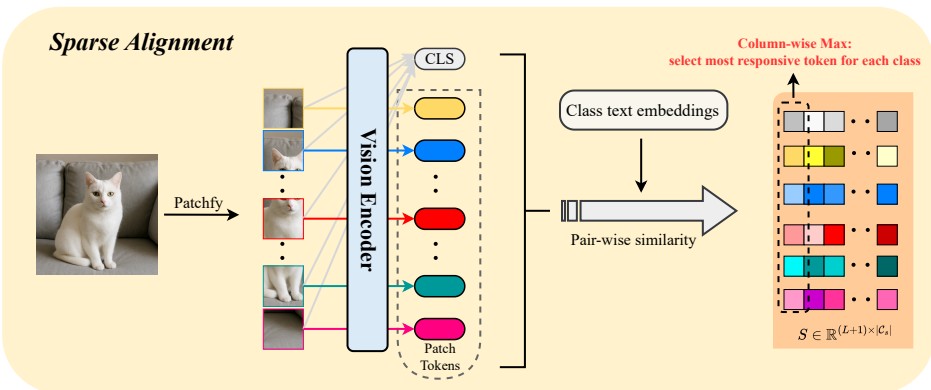

Figure 7: Pipeline of sparse alignment.

## B  MORE ABLATION EXPERIMENTS

**More Comparison with SOTA Methods.** Due to space limitations, we report here a more comprehensive comparison of experiments, in which we additionally include more impressive CLIP-based methods.

Table 8: The experimental results on closed-world settings.

| Method | UT-Zappos | | | | MIT-States | | | | C-GQA | | | |
|---|---|---|---|---|---|---|---|---|---|---|---|---|
| | S | U | HM | AUC | S | U | HM | AUC | S | U | HM | AUC |
| CLIP[ICML'21] (Radford et al., 2021) | 15.8 | 49.1 | 15.6 | 5.0 | 30.2 | 46.0 | 26.1 | 11.0 | 7.5 | 25.0 | 8.6 | 1.4 |
| CoOp[IJCV'22] (Zhou et al., 2022) | 52.1 | 49.3 | 34.6 | 18.8 | 34.4 | 47.6 | 29.8 | 13.5 | 20.5 | 26.8 | 17.1 | 4.4 |
| PCVL[Arxiv'22] (Xu et al., 2022) | 64.4 | 64.0 | 46.1 | 32.2 | 48.5 | 47.2 | 35.3 | 18.3 | - | - | - | - |
| HPL[IJCAI'23] (Wang et al., 2023a) | 63.0 | 68.8 | 48.2 | 35.0 | 47.5 | 50.6 | 37.3 | 20.2 | 30.8 | 28.4 | 22.4 | 7.2 |
| CSP[ICLR'23] (Nayak et al., 2023) | 64.2 | 66.2 | 46.6 | 33.0 | 46.6 | 49.9 | 36.3 | 19.4 | 28.8 | 26.8 | 20.6 | 6.2 |
| DFSP[CVPR'23] (Lu et al., 2023) | 66.7 | 71.7 | 47.2 | 36.0 | 46.9 | 52.0 | 37.3 | 20.6 | 38.2 | 32.0 | 27.1 | 10.5 |
| PLID[ECCV'24] (Bao et al., 2023) | 67.3 | 68.8 | 52.4 | 38.7 | 49.7 | 52.4 | 39.0 | 22.1 | 38.8 | 33.0 | 27.9 | 11.0 |
| CDS[CVPR'24] (Li et al., 2024) | 63.9 | 74.8 | 52.7 | 39.5 | 50.3 | 52.9 | 39.2 | 22.4 | 38.3 | 34.2 | 28.1 | 11.1 |
| Troika[CVPR'24] (Huang et al., 2024) | 66.8 | 73.8 | 54.6 | 41.7 | 49.0 | 53.0 | 39.3 | 22.1 | 41.0 | 35.7 | 29.4 | 12.4 |
| CAILA[WACV'24] (Zheng et al., 2024) | 67.8 | 74.0 | 57.0 | 44.1 | 51.0 | 53.9 | 39.9 | 23.4 | 43.9 | 38.5 | 32.7 | 14.8 |
| RAPR[AAAI'24] (Jing et al., 2024) | 69.4 | 72.8 | 56.5 | 44.5 | 50.0 | 53.3 | 39.2 | 22.5 | 45.6 | 36.0 | 32.0 | 14.4 |
| LogiCzsl[CVPR'25] (Wu et al., 2025) | 69.6 | 74.9 | 57.8 | 45.8 | 50.8 | 53.9 | 40.5 | 23.4 | 44.4 | 39.4 | 33.3 | 15.3 |
| ClusPro[ICLR'25] (Qu et al., 2025) | 70.7 | 76.0 | 58.5 | 46.6 | 52.1 | **54.0** | 40.7 | 23.8 | 44.3 | 37.8 | 32.8 | 14.9 |
| SAC | **73.3** | **76.8** | **62.0** | **50.0** | **53.2** | 53.0 | **40.8** | **24.0** | **45.8** | **39.5** | **34.8** | **16.2** |

**More Ablation Study on Hyper-Parameteres.** We further study the impact of hyper-parameters on performance, including weight coefficient $\alpha$ in distillation loss Eq. 7, weight coefficient $\beta$, $\gamma$ in inference Eq. 12 and number of blocks $K$ in VAC.

**Influence of Loss Coefficient Weight $\alpha$.** First, we conduct experiments on $\alpha$ to investigate the impact of the distillation loss in Eq. 8 on the Visual Adaptive Condensation module and the results are reported in Fig. 8. According to the analysis, we set the $\alpha$ as 0.5, 0.9 and 0.5 for UT-Zappos, MIT-States and C-GQA, respectively. As we can see, the distillation loss provides a clear performance gain for the VAC module. Ablating this loss (*e.g.*, setting the weight to 0) reduces the VAC objective to a standard classification loss, resulting in notably poorer performance.

**Influence of Inference Weight of $\beta$ and $\gamma$.** Then, we conduct experiments on inference weight $\beta$ and $\gamma$ in Eq. 12 and the results are reported in Fig. 9 and Fig. 10, respectively. We observe that the optimal parameter settings differ across benchmarks. We hypothesize that this arises from varying dataset characteristics, including differences in object or attribute contamination from surrounding regions. Consequently, adjusting the contribution of our modules yields different levels of perfor-

Table 9: The experimental results on open-world settings.

| Method | UT-Zappos | | | | MIT-States | | | | C-GQA | | | |
|---|---|---|---|---|---|---|---|---|---|---|---|---|
| | S | U | HM | AUC | S | U | HM | AUC | S | U | HM | AUC |
| CLIP[ICML'21] (Radford et al., 2021) | 15.7 | 20.6 | 11.2 | 2.2 | 30.1 | 14.3 | 12.8 | 3.0 | 7.5 | 4.6 | 4.0 | 0.3 |
| CoOp[IJCV'22] (Zhou et al., 2022) | 52.1 | 31.5 | 28.9 | 13.2 | 34.6 | 9.3 | 12.3 | 2.8 | 21.0 | 4.6 | 5.5 | 0.7 |
| PCVL[Arxiv'22] (Xu et al., 2022) | 64.6 | 44.0 | 37.1 | 21.6 | 48.5 | 16.0 | 17.7 | 6.1 | - | - | - | - |
| HPL[IJCAI'23] (Wang et al., 2023a) | 63.4 | 48.1 | 40.2 | 24.6 | 46.4 | 18.9 | 19.8 | 6.9 | 30.1 | 5.8 | 7.5 | 1.4 |
| CSP[ICLR'23] (Nayak et al., 2023) | 64.1 | 44.1 | 38.9 | 22.7 | 46.3 | 15.7 | 17.4 | 5.7 | 28.7 | 5.2 | 6.9 | 1.2 |
| DFSP[CVPR'23] (Lu et al., 2023) | 66.8 | 60.0 | 44.0 | 30.3 | 47.5 | 18.5 | 19.3 | 6.8 | 38.3 | 7.2 | 10.4 | 2.4 |
| PLID[ECCV'24] (Bao et al., 2023) | 67.6 | 55.5 | 46.6 | 30.8 | 49.1 | 18.7 | 20.0 | 7.3 | 39.1 | 7.5 | 10.6 | 2.5 |
| CDS[CVPR'24] (Li et al., 2024) | 64.7 | 61.3 | 48.2 | 32.3 | 49.4 | 21.8 | 22.1 | 8.5 | 37.6 | 8.2 | 11.6 | 2.7 |
| Troika[CVPR'24] (Huang et al., 2024) | 66.4 | 61.2 | 47.8 | 33.0 | 48.8 | 18.7 | 20.1 | 7.2 | 40.8 | 7.9 | 10.9 | 2.7 |
| CAILA[WACV'24] (Zheng et al., 2024) | 67.8 | 59.7 | 49.4 | 32.8 | 51.0 | 20.2 | 21.6 | 8.2 | 43.9 | 8.0 | 11.5 | 3.1 |
| RAPR[AAAI'24] (Jing et al., 2024) | 69.4 | 59.4 | 47.9 | 33.3 | 49.9 | 20.1 | 21.8 | 8.2 | 45.5 | 11.2 | 14.6 | 4.4 |
| LogiCzsl[CVPR'25] (Wu et al., 2025) | 69.6 | 63.7 | 50.8 | 36.2 | 50.7 | 21.4 | 22.4 | 8.7 | 43.7 | 9.3 | 12.6 | 3.4 |
| ClusPro[ICLR'25] (Qu et al., 2025) | 71.0 | 66.2 | 54.1 | 39.5 | 51.2 | **22.1** | 23.0 | 9.3 | 41.6 | 8.3 | 11.6 | 3.0 |
| **SAC** | **72.9** | **66.7** | **54.8** | **42.3** | **52.9** | 21.2 | **23.1** | **9.4** | **45.5** | 11.5 | **15.3** | **4.6** |

mance gain. Therefore, based on our experimental results, we set $\beta$ as 0.3, 0.7 and 0.7, and set $\gamma$ as 0.5, 0.4 and 0.1 for UT-Zappos, MIT-States and C-GQA, respectively.

**Influence of Number of Blocks in VAC.** In addition, we report the ablation study for $K$, number of blocks in *Visual Adaptive Condensation* module. After a comprehensive evaluation, we ultimately set $K$ as 3, 3 and 1 for UT-zappos, MIT-States and C-GQA, respectively. The detailed performance are reported in Tab. 10.

**Influence of Number of Stored Samples in Memory Bank.** We empirically investigate the impact of the number of stored samples per composition in the memory bank. As illustrated in Tab. 11, we observe that a small memory size leads to suboptimal and unstable performance due to limited sample diversity, and the performance becomes consistent as the memory size increases. However, continually increasing the memory size (*e.g.*, by initializing new slots as zero vectors) may dilute retrieval weights in Eq. 10. Based on this analysis, we set the number of samples to 16, 24, and 16 for datasets UT-Zappos, MIT-States, and C-GQA, respectively, and set the temperature $\tau_{mb}$ as 0.1 in Eq. 10 to sharpen the weight of effective samples.

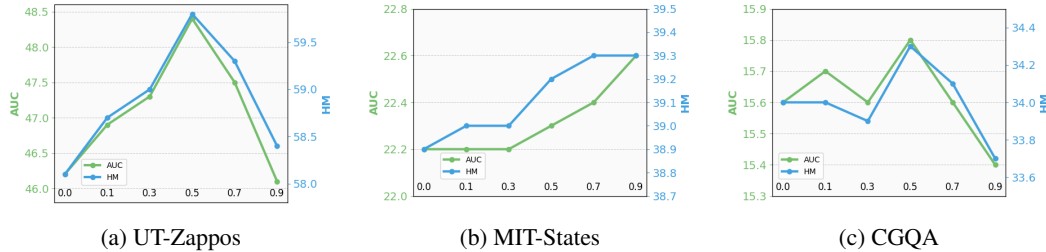

| (a) UT-Zappos | (b) MIT-States | (c) CGQA |
|---|---|---|

Figure 8: Impact of $\alpha$ across three datasets.

## C MORE QUALITATIVE EXPERIMENTS

**More Qualitative Results.** Here, we report more qualitative results in UT-Zappos, MIT-States and C-GQA datasets. As shown in Fig. C, our method can predict accurate results where the baseline makes mistakes. For example, baseline is struggle to distinguish similar objects, *e.g.*, "*countertop*" and "*drawer*", "*box*" and "*cooler*". Meanwhile, without filtering redundant information, baseline is misled by extraneous visual content, *e.g.*, baseline focuses on object "*iron fence*", not "*calm water*". These results demonstrate the effectiveness of our method: by suppressing redundant information, our method is able to make more accurate predictions.

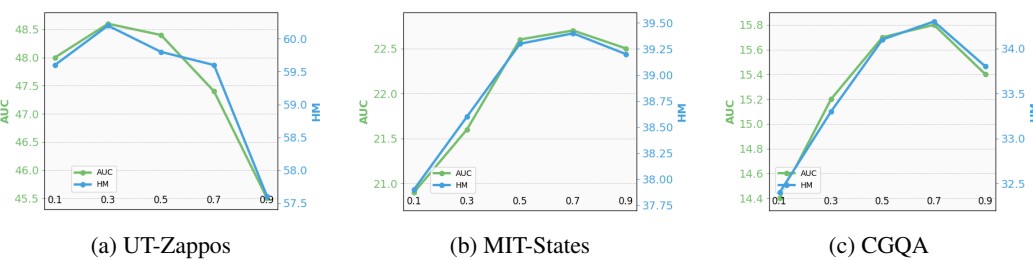

(a) UT-Zappos     (b) MIT-States     (c) CGQA

Figure 9: Impact of $\beta$ across three datasets.

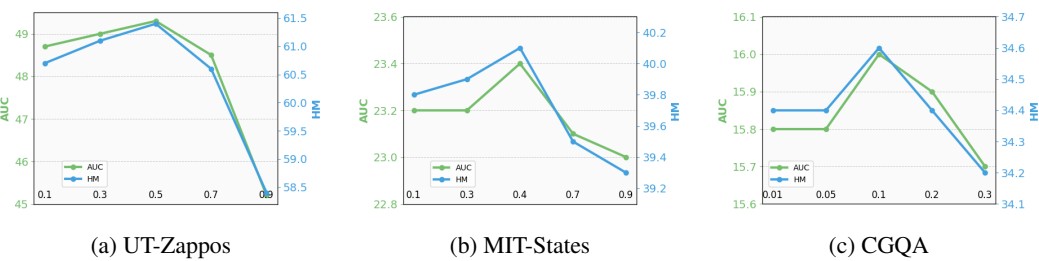

(a) UT-Zappos     (b) MIT-States     (c) CGQA

Figure 10: Impact of $\gamma$ across three datasets.

**More Visualization Results.** As shown in Fig. 12, we present more visualization results of *Visual Adaptive Condensation* (VAC) module in C-GQA dataset. We can observe that our proposed VAC is capable of excavating critical visual information without disturbing by redundant visual cues, such as, "*bear*" in "*green leaf*", "*wall*" in "*mess fence*" and "*cat*" in "*gray seat*", where the main objects are more salient and occupy greater space. These results demonstrate the effectiveness of our proposed VAC.

Table 10: Impact of $K$ in VAC across three datasets.

| (a) UT-Zappos | | | | | (b) MIT-States | | | | | (c) C-GQA | | | |
|---|---|---|---|---|---|---|---|---|---|---|---|---|---|
| | **S** | **U** | **HM** | **AUC** | | **S** | **U** | **HM** | **AUC** | | **S** | **U** | **HM** | **AUC** |
| K=1 | 67.6 | 74.1 | 57.0 | 43.7 | $K$=1 | 50.3 | 52.1 | 38.6 | 22.0 | **K=1** | **45.6** | 38.6 | **34.1** | **15.7** |
| **K=3** | **71.2** | **76.2** | **58.9** | **46.7** | **K=3** | **50.6** | **52.1** | **39.3** | **22.4** | $K$=2 | 45.2 | **39.1** | 33.9 | 15.7 |
| K=5 | 69.5 | 76.0 | 57.7 | 45.5 | $K$=5 | 50.6 | 51.9 | 39.1 | 22.2 | $K$=3 | 45.7 | 37.8 | 33.6 | 15.4 |

## D    PSEUDO-CODE

**Training Scheme for SAC.** In this section, we provide a detailed training scheme for our proposed SAC framework, which can be divided into three stages. **Stage I: Sparse Alignment**, we conduct sparse alignment between textual representations and patch visual representations. Leveraging this information-balanced training data, we optimize LoRA (Zanella & Ben Ayed, 2024) for the visual encoder in CLIP. **Stage II: Visual Adaptive Condensation**, with the reduced visual information in the above alignment, the module is guided to adaptively excavate critical visual information within the image, which preserves potential discarded yet valuable information in stage I. **Stage III: Dynamically Updated Memory Bank**, we first initialize memory bank through training data and dynamicall update the memory bank during inference.

## E    STATEMENT FOR USING LARGE LANGUAGE MODELS.

In this section, we illustrate the contributions of author contributions and LLMs tools.

Table 11: Impact of $N$ in memory bank across three datasets.

| (a) UT-Zappos | S | U | HM | AUC |
|---|---|---|---|---|
| N=2 | 72.3 | 76.2 | 61.1 | 48.6 |
| N=4 | 72.6 | 76.2 | 61.1 | 48.8 |
| N=8 | 72.3 | 76.2 | 60.9 | 48.8 |
| **N=16** | **73.1** | **76.2** | **61.0** | **49.2** |
| N=24 | 72.9 | 76.2 | 60.9 | 49.0 |

| (b) MIT-States | S | U | HM | AUC |
|---|---|---|---|---|
| N=4 | 50.5 | 52.6 | 38.7 | 22.2 |
| N=8 | 51.4 | 52.6 | 39.3 | 22.6 |
| N=16 | 51.5 | 52.6 | 39.3 | 22.6 |
| **N=24** | **51.9** | **52.6** | **39.2** | **22.7** |
| N=24 | 51.9 | 52.6 | 39.0 | 22.7 |

| (c) C-GQA | S | U | HM | AUC |
|---|---|---|---|---|
| N=2 | 43.9 | 38.8 | 33.5 | 15.2 |
| N=4 | 44.7 | 38.8 | 33.6 | 15.4 |
| N=8 | 44.9 | 38.8 | 33.9 | 15.5 |
| **N=16** | **45.1** | **38.8** | **34.0** | **15.6** |
| N=24 | 45.0 | 38.8 | 34.0 | 15.6 |

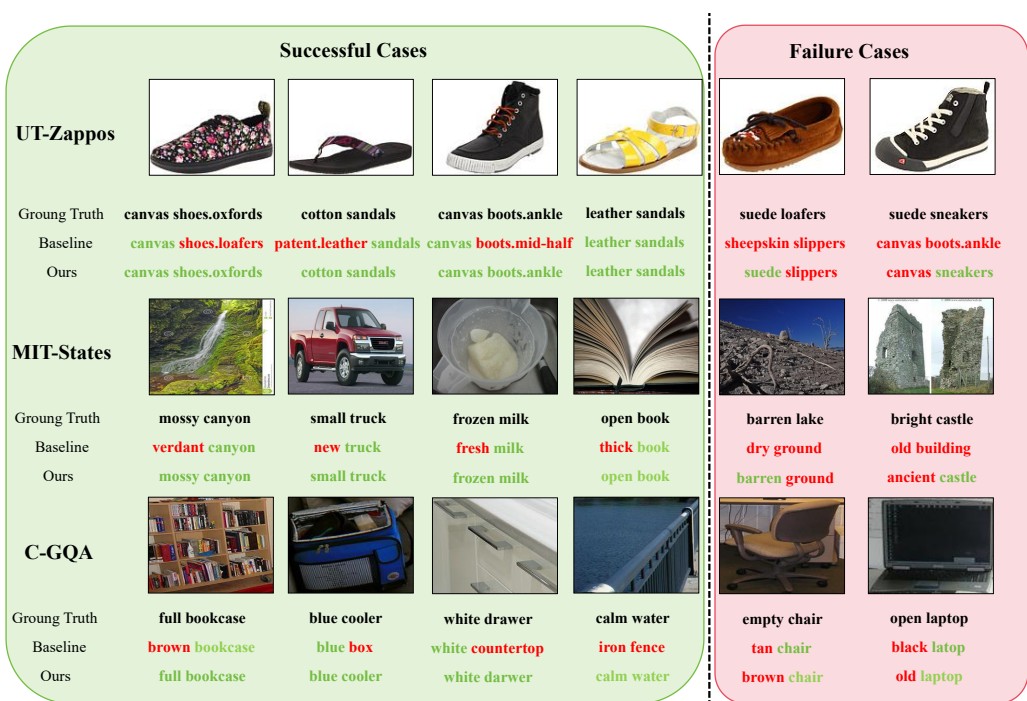

Figure 11: More qualitative results of our method on three datasets.

**Core Contributions** (by the authors): Conception of the Sparse Alignment idea, design the architecture and loss function for modules, design and implementation of all experiments, data analysis, and interpretation of all results.

**Assistance from LLMs**: In the final stages of manuscript preparation, we used AI tools (ChatGPT/GPT-4 and DeepSeek) for specific, non-intellectual tasks to improve presentation quality. Their use was strictly limited to: 1) Language Polishing: Identifying and correcting typographical, grammatical, and spelling errors. 2) Syntax and Style: Rephrasing sentences for improved readability and academic tone, without altering technical meaning. 3) LaTeX Code Debugging: Ensuring consistency in reference formatting, figure/table labels, and other LaTeX conventions.

The models did not contribute to the scientific ideas, experimental design, or conclusions of this work. The authors reviewed and edited all AI-suggested changes and assume full responsibility for the published content.

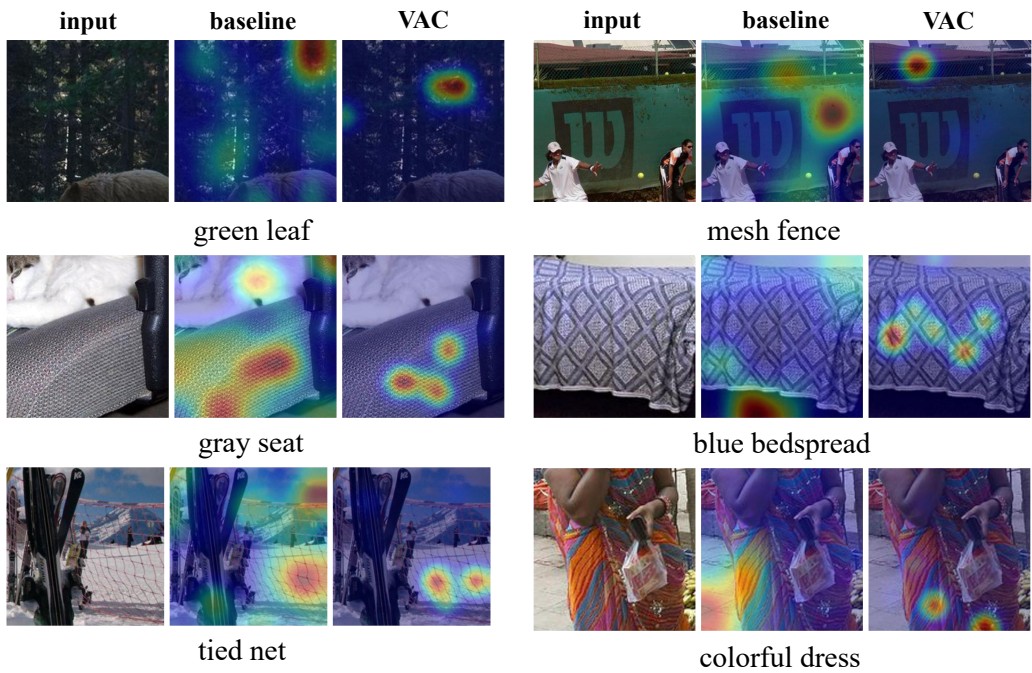

Figure 12: More visualization results of VAC module in C-GQA dataset.

---

**Algorithm 1** Training Scheme for SAC.

---

**Input:** training data $\mathcal{D}_{tr}$, visual encoder of CLIP $\phi_{\text{vis}}$, textual encoder of CLIP $\psi_{\text{txt}}$,
learnable soft prompts $\boldsymbol{\theta}_t = [\boldsymbol{\theta}_a, \boldsymbol{\theta}_o, \boldsymbol{\theta}_c]$, visual adaptive condensation module $\boldsymbol{\theta}_{vac}$,
LoRA weight $\boldsymbol{\theta}_{LoRA}$, memory bank $\mathbf{B}$.
**Output:** optimized: LoRA weight $\boldsymbol{\theta}_{LoRA}$, learnable soft prompts $\boldsymbol{\theta}_t = [\boldsymbol{\theta}_a, \boldsymbol{\theta}_o, \boldsymbol{\theta}_c]$,
visual adaptive condensation module $\boldsymbol{\theta}_{vac}$; updated memory bank $\mathbf{B}$.
1: **Stage I:**, randomly initialize parameters $\boldsymbol{\theta}_{LoRA}$; load pre-trained parameters visual encoder of
CLIP $\phi_{\text{vis}}$, textual encoder of CLIP$\psi_{\text{txt}}$, learnable soft prompts $[\boldsymbol{\theta}_a, \boldsymbol{\theta}_o, \boldsymbol{\theta}_c]$.
2: **while** *not converged* **do**
3:     batch of training data $(\mathcal{X}_b, \mathcal{Y}_b)$
4:     conducting sparse alignment by visual reduction in Eq. 2
5:     calculating basic learning objective $\mathcal{L}_{base}$ in Eq. 4
6:     optimize parameters $\boldsymbol{\theta}\,(\boldsymbol{\theta}_{LoRA}, \boldsymbol{\theta}_t) = \boldsymbol{\theta} - \nabla_{\boldsymbol{\theta}}(\mathcal{L}_{base}(\mathcal{X}_b, \mathcal{Y}_b; \boldsymbol{\theta}))$
7: **end while**
8: **Stage II:** randomly initialize parameters $\boldsymbol{\theta}_{VAC}$.
9: **while** *not converged* **do**
10:     batch of training data $(\mathcal{X}_b, \mathcal{Y}_b)$
11:     condense visual information within image into $\boldsymbol{v}_q$
12:     calculation prediction $\boldsymbol{p}_{vac}$ of VAC by Eq. 5 and $\boldsymbol{p}_{sa}$ of SA by Eq. 3
13:     calculating learning objective $\mathcal{L}_{base}^{vac}$ in Eq. 6 and $\mathcal{L}_{kl}$ in Eq. 7
14:     optimize parameters $\boldsymbol{\theta}\,(\boldsymbol{\theta}_{vac}) = \boldsymbol{\theta} - \nabla_{\boldsymbol{\theta}}((1 - \alpha) \cdot \mathcal{L}_{base}^{vac}(\mathcal{X}_b, \mathcal{Y}_b; \boldsymbol{\theta}) + \alpha \cdot \mathcal{L}_{kl}(\mathcal{X}_b, \mathcal{Y}_b; \boldsymbol{\theta}))$
15: **end while**
16: **Stage III:** initialize stored samples for seen compositions in memory bank $\mathbf{B}$ by Eq. 9.
17: **for** batch of testing data $\mathcal{X}_b$ **do**
18:     calculating predictions $\boldsymbol{p}_{sa}$, $\boldsymbol{p}_{vac}$ and $\boldsymbol{p}_{bank}$ from three modules by Eq. 3, Eq. 5 and Eq. 10,
respectively
19:     obtain final prediction by Eq. 12
20:     utilizing $\boldsymbol{p}_{vac}$ to update memory bank by Eq. 9
21: **end for**
22: calculating the results of each evaluation metric with the final prediction
23: **return** optimized LoRA weight $\boldsymbol{\theta}_{LoRA}$, learnable soft prompts $\boldsymbol{\theta}_t = [\boldsymbol{\theta}_a, \boldsymbol{\theta}_o, \boldsymbol{\theta}_c]$,
visual adaptive condensation module $\boldsymbol{\theta}_{vac}$; updated memory bank $\mathbf{B}$.

---

