# OpenReview forum: "One Patch, One Text: Sparse Alignment for Closing CLIP's Modality Gap for Compositional Zero-Shot Learning"
_ICLR.cc/2026/Conference — ICLR 2026 Conference Withdrawn Submission_

### Official Review · Reviewer_yrZ4 · 2025-10-25

**Soundness:** 2
**Presentation:** 3
**Contribution:** 2
**Rating:** 4
**Confidence:** 3

**Summary:**

This paper addresses the problem of Compositional Zero-Shot Learning (CZSL) , focusing on a key limitation of CLIP-based models: the modality gap. The authors hypothesize that this gap stems from an "information imbalance" in the training data, where visually-dense images (encoded by many patch tokens) are aligned with information-sparse text descriptions. The standard [CLS] token, by aggregating all visual information, introduces redundant context that weakens the alignment.

**Strengths:**

1. The paper identifies a critical and plausible root cause for the modality gap in CZSL: the information imbalance between rich visual inputs and sparse textual labels. The pilot experiments, which show that even naive random token dropping can improve performance and reduce the gap, provide a solid foundation for the paper's core hypothesis.

2. The central idea of SA is novel and elegant. Shifting the alignment objective from the "polluted" global [CLS] token to the single, most-relevant patch token is an intuitive and principled method to enforce information balance and mitigate the gap.

3. The method achieves state-of-the-art results across three challenging datasets , demonstrating significant improvements in both closed-world (Table 2) and open-world (Table 3) settings.

**Weaknesses:**

1. Contradictory Motivation (SA vs. VAC): The paper is built on the premise that aggregating information from all patches (like the [CLS] token) is detrimental. However, Stage II (VAC) is then introduced to do exactly that: aggregate information from all patch tokens. What’s the difference between VAC and traditional CLS token of CLIP obtained by Attention Pooling aside from the distillation guidance.

2. Unfair Comparison: The Stage III memory bank is "dynamically updated during inference" using test samples. This is a form of Test-Time Augmentation. Comparing SAC's results (Tables 2 & 3) against SOTA methods that presumably without test time augmentation is unfair.

**Questions:**

1. Overly Complex Framework: The proposed SAC method is a highly complex, multi-stage pipeline. It combines sparse alignment training, a separate condensation module (VAC), knowledge distillation, and a dynamic k-NN memory bank with TTA.

2. No comparison of visualization results between SA and VAC.

---

### Official Review · Reviewer_Weqx · 2025-10-30

**Soundness:** 3
**Presentation:** 3
**Contribution:** 2
**Rating:** 4
**Confidence:** 4

**Summary:**

This paper investigates the modality gap problem in Compositional Zero-Shot Learning (CZSL) with CLIP. The authors argue that this phenomenon stems from an imbalance between redundant information in the image modality and sparse information in the text modality. To address this, they propose a three-stage framework called SAC (Sparse Alignment for Closing Modality Gap), consisting of:
1. Sparse Alignment (SA): Aligns image and text features at the patch level, reducing visual redundancy through sparse selection;
2. Visual Adaptive Condensation (VAC): Further condenses key visual features to prevent over-sparsification;
3. Dynamic Memory Bank (MB): Dynamically updates visual representations during inference to enhance model generalization.
Experiments on UT-Zappos, MIT-States, and C-GQA datasets show that SAC outperforms several existing methods under both closed- and open-set CZSL settings.
Overall, the paper’s motivation is clear and structure well-organized. Experiments are relatively sufficient, but the work lacks strong novelty and theoretical depth, with some heuristic design choices.

**Strengths:**

1.	Accurate problem formulation: The modality gap indeed exists in current CLIP models, and the authors quantify it using a Relative Modality Gap metric.
2.	Simple and effective method: The SAC module is lightweight, plug-and-play, and can be seamlessly integrated into existing CLIP-based CZSL frameworks, consistently improving performance on multiple benchmarks.

**Weaknesses:**

1.  There is no rigorous mathematical proof of how sparse selection reduces modality discrepancy. The explanation remains empirical.
2. The core method mainly combines existing alignment and sparsity mechanisms, lacking new learning objectives or structural innovations.
3.  Although results on three benchmarks are reported, there is no direct comparison with more recent open-vocabulary or Vision-LLM methods (e.g., OV-CZSL, VisionLLM-C). The ablation study also fails to adequately analyze sensitivity to sparsity rate or thresholds.
4.  The dynamic update rule is heuristic, and the potential for drift or error accumulation during long-sequence inference is not discussed.

**Questions:**

1. In the sparse selection strategy, have you considered soft attention or learnable gating mechanisms to prevent excessive information loss?
2. Could the dynamic memory bank’s inference-time update cause potential data leakage or overfitting?
3. Can this approach generalize to open-vocabulary detection or retrieval tasks?
4. Have you explored combining SAC with LoRA or Prompt-tuning lightweight fine-tuning techniques?

---

### Official Review · Reviewer_8YZT · 2025-10-31

**Soundness:** 2
**Presentation:** 2
**Contribution:** 2
**Rating:** 4
**Confidence:** 5

**Summary:**

This paper tackles Compositional Zero-Shot Learning (CZSL) by addressing the modality gap between visual and text representations in CLIP. The proposed SAC (Sparse Alignment for Closing modality gap) mitigates this gap from two perspectives: reducing cross-modal information imbalance and partially bypassing cross-modal matching for certain samples.
SAC includes three stages: (1) Sparse Alignment, which aligns only the most relevant image patches with text to reduce redundancy. (2)Visual Adaptive Condensation, which aggregates key visual cues into a compact embedding via attention and  distillation from Sparse Alignment. (3)Dynamic Memory Bank, wwhich stores and dynamically updates visual prototypes for both seen and unseen compositions during training and inference. Experiments on MIT-States, UT-Zappos, and C-GQA show that SAC achieves strong or state-of-the-art performance, effectively narrowing the modality gap and enhancing compositional generalization.

**Strengths:**

1. The paper approaches the CZSL problem from a novel perspective, tackling the modality gap in CLIP through sparse alignment and adaptive visual condensation, and achieves state-of-the-art performance across multiple benchmarks.

1. The experimental analysis is thorough and convincing, including extensive ablations on each module and hyperparameter choice, which clearly demonstrate the contribution of each component.

**Weaknesses:**

1.  The use of a Dynamic Memory Bank during testing raises methodological concerns. It effectively leverages test-set statistics, which violates the inductive learning setting. Moreover, the incremental update process means that test results can depend on the order of input samples, reducing reproducibility.

2. The paper’s main claim is that reducing redundant visual patches helps close the modality gap and improves CZSL performance. However, the analysis of the modality gap is limited to the C-GQA dataset and remains correlational. There is no direct evidence showing that closing the gap causally leads to better results across datasets.

3. The presentation lacks clarity in several places. For instance, Equation (11) could mislead readers into thinking the SA and VAC modules are trained jointly, whereas the appendix clarifies that they are trained in order. The explanation of Figure 4 (right) is vague and could benefit from more detail.

**Questions:**

1. Please provide more comprehensive evidence of the relationship between the modality gap and CZSL performance. For instance, can you report the modality-gap metric (e.g., RMG) and accuracy across all three datasets, and, if possible, compare them with previous CLIP-based methods to show consistency of the observed trend?

2. On C-GQA, could the improvements from Sparse Alignment result mainly from filtering background noise or reducing overfitting rather than decreasing the information imblance of different modality?

3. Prior work [1] reports that a larger modality gap shows only a mild positive correlation with downstream performance, "A larger modality gap has mild positive correlation with downstream performance. However, there is no indication that a larger modality gap leads to a better performance;rather, it suggests the presence of common confounders (e.g., model size)" . While Figure 2 in this paper shows a clear negative correlation between AUC and RMG. Please clarify this difference and provide further analysis. In addition, Figure 2 shows that small random-drop ratios (0.1%–0.5%) reduce the gap, but a larger drop (10%) increases it—please explain this  trend and whether it is related to information loss, overfitting reduction, or dataset-specific background effects.

[1] Simon Schrodi, David T Hoffmann, Max Argus, Volker Fischer, and Thomas Brox. Two effects, one trigger: On the modality gap, object bias, and information imbalance in contrastive visionlanguage models. In The Thirteenth International Conference on Learning Representations, 2025.

---

### Official Review · Reviewer_Cco9 · 2025-10-31

**Soundness:** 3
**Presentation:** 2
**Contribution:** 2
**Rating:** 4
**Confidence:** 3

**Summary:**

The paper proposes a three-stage framework called Sparse Alignment for Closing modality gap (SAC) to address the "modality gap" in CLIP for Compositional Zero-Shot Learning (CZSL). This paper believes the modality gap is mainly caused by information imbalance between rich visual data and sparse text descriptions.
Instead of using the standard [CLS] token, this work aligns each text prompt only with its single most semantically relevant visual patch token. The VAC module uses a learnable query to adaptively aggregate critical visual information from all patch tokens into a condensed representation. A memory bank is created to store high-confidence condensed visual representations from the VAC module for all compositions.

**Strengths:**

1. The method targets the "modality gap", which is a challenge of CLIP-based models.
2. The method was well-designed, and the experimental results were competitive.

**Weaknesses:**

1. This method fails to offer a targeted solution to the fundamental challenge of CZSL when attributes and objects are "heavy entanglement".
2. This method only applies to CLIP-based models, thus having limitations. Does the "modality gap" also exist in non-CLIP-based models? Can this method also address it? The paper lacks discussion on this aspect.
3. The memory bank in Phase III needs to be "continuously updated" during inference. This means that the model is not static at test time, which increases the computational overhead of inference.
4. The performance of this method is sensitive to multiple sets of hyperparameters, which need to be specifically tuned for different datasets, which reduces the model's general applicability. Figures 8, 9, 10 and Table 11 in the appendix show that if these hyperparameters are not set properly, the model's performance (AUC and HM) will decrease significantly.
5. The memory bank is continuously updated with test samples during the inference process. This means that the model's predictive performance on a later test sample benefits from its predictions on a previous test sample, which means the model is sensitive to the order of the test samples. This paper does not investigate the impact of the order of the test sets on the results.
6. The “Baseline” or “Full” model used for comparison is a “clip fine-tuned with LoRA”. The significant performance improvement shown in the paper is based on a strong, fine-tuned baseline that has been adapted to downstream tasks and may have partially alleviated the modal gap. The authors did not show ablation experiments with LoRA removed.

**Questions:**

See the weaknesses.

---

### Note · Authors · 2025-11-13

I have read and agree with the venue's withdrawal policy on behalf of myself and my co-authors.